

# Usability and acceptance of crowd-based early warning of harmful algal blooms

Lindung Parningotan Manik[1,2], Hatim Albasri[3], Reny Puspasari[3], Aris Yaman[4], Shidiq Al Hakim[2], Al Hafiz Akbar Maulana Siagian[2], Siti Kania Kushadiani[2], Slamet Riyanto[2], Foni Agus Setiawan[2], Lolita Thesiana[3], Meuthia Aula Jabbar[5], Ramadhona Saville[6] and  Masaaki Wada[7]

[1] Faculty of Information Technology, University of Nusa Mandiri, Jakarta, Indonesia
[2] Research Center for Data and Information Sciences, National Research and Innovation Agency, Bandung, Indonesia
[3] Research Center for Fisheries, National Research and Innovation Agency, Jakarta, Indonesia
[4] Research Center for Computing, National Research and Innovation Agency, Bogor, Indonesia
[5] Department of Aquatic Resources Management, Jakarta Technical University of Fisheries, Jakarta, Indonesia
[6] Department of Agribusiness Management, Tokyo University of Agriculture, Tokyo, Japan
[7] School of Systems Information Science, Future University Hakodate, Hokkaido, Japan

## ABSTRACT

Crowdsensing has become an alternative solution to physical sensors and apparatuses. Utilizing citizen science communities is undoubtedly a much cheaper solution. However, similar to other participatory-based applications, the willingness of community members to be actively involved is paramount to the success of implementation. This research investigated factors that affect the continual use intention of a crowd-based early warning system (CBEWS) to mitigate harmful algal blooms (HABs). This study applied the partial least square-structural equation modeling (PLS-SEM) using an augmented technology acceptance model (TAM). In addition to the native TAM variables, such as perceived ease of use and usefulness as well as attitude, other factors, including awareness, social influence, and reward, were also studied. Furthermore, the usability factor was examined, specifically using the System Usability Scale (SUS) score as a determinant. Results showed that usability positively affected the perceived ease of use. Moreover, perceived usefulness and awareness influenced users' attitudes toward using CBEWS. Meanwhile, the reward had no significant effects on continual use intention.

# INTRODUCTION

The aquaculture subsector has proliferated in the last three decades, contributing 46% of the total fish production according to *FAO (2020)*. However, several emerging challenges affect the development of the fish farming industry, including harmful algae blooms (HABs), an aquatic environmental event caused by excessive growth of certain species of phytoplankton/algae. A massive HAB event could lead to mass mortality of higher trophic marine organisms within a large geographical area, including farmed fish. The

Corresponding author
Lindung Parningotan Manik,
lind008@brin.go.id

damaging effect of HABs is caused by toxins released by HAB causative species and oxygen depletion in the water column, leading to asphyxiation in fish. Frequent HAB events could markedly reduce the economic capacity of a mariculture-dependent coastal region due to its unpredictability, scale, and high fish mortality rate. For example, according to a study by *León-Muñoz et al. (2018)*, large fish farms in Chile reported a loss of nearly 40 thousand tonnes of cultured salmon during re-occurrences of HAB events between 2015 and 2016. In Indonesia, HABs have occurred regularly in Lampung Bay. In 2018, small-scale fish farmers lost at least 30 tonnes of farmed fish due to a single HAB event, according to a study by *Puspasari et al. (2018)*. Both cases have caused long-term social, economic, political, and environmental disruptions in Chile and Indonesia. Although HABs' negative impacts have severely affected mariculture, capture fisheries, and human health, early warning system (EWS) to detect and mitigate these adverse effects are rarely investigated.

Studies by *Yuan et al. (2018)* and *Davidson et al. (2021)* have developed and integrated various EWSs into fish farming activities to convey environmental conditions in (near) real-time, such as water or weather quality EWSs. For example, a water quality EWS automatically senses a poor water condition and warns farmers and other related parties of the situation. This system remotely collects in-situ data of various water quality parameters using optical or combined optical-chemical sensor equipment installed at specific sites. The data are then typically processed to filter out error/outliers data readings, interpolate missing data, and categorize the data and dataset based on predefined classes of water quality conditions for specific farmed fish species and/or farming systems. The processed information, usually in a more straightforward format, is then communicated to the user *via* various visual displays to aid the users in decision-making in response to the changing condition. An EWS requires as many consistent, complete, and continuous datasets as possible to generate accurate alerts. Collecting data for the EWS is generally carried out through sensors deployed in the monitored water area. The challenge is that data's increasing amount, type, and spatiotemporal coverage is expensive and requires many apparatuses. In order to overcome this, another mechanism in data collection can be used, namely crowdsourcing, *i.e.,* the new online distributed production model in which people collaborate and may be awarded to complete a task.

Crowdsourcing technology has been implemented in various use cases, including biodiversity contexts. For example, iNaturalist, developed by *Aristeidou et al. (2021)*, facilitates global citizen scientists to record and share observations of plants and animals. Other researchers have also studied the implementation of crowdsourcing, such as *Sullivan et al. (2009)* in collecting bird observation data in eBird and *Zhou et al. (2018)* in collecting images of plant phenomics. Specifically, for the EWS context, HABscope was developed by *Hardison et al. (2019)* as a tool to help with early warning of respiratory irritation caused by harmful blooms. Furthermore, *Mishra et al. (2020)* developed CyanoTRACKER to observe cyanobacterial blooms globally using a cloud-based integrated multi-platform architecture. Inspired by these studies and the increasing occurrences of HABs in Indonesia's coastal waters, since 2019, Alboom has been developed as a crowdsourcing application used by citizen scientists to record, store, analyze, share, and provide early warning information regarding HABs.

Individuals use the Alboom mobile application to collect geotagged images and report visual information regarding water quality and weather conditions in their locality, whether there are HABs or not. Non-HAB data are intended to provide baseline information for the "normal situation" in the areas of interest or serve as a "precursor" condition if HABs occur. In contrast, HAB data and visual information are used to validate HAB events and later as data sources for HAB early warning information for the local community as well as regional and national mitigation of HABs. In addition, Algies, an expert system, has also been developed by *Setiawan et al. (2021)* using an ontology of algae to speed up the identification process of algae that causes HABs. Alboom and Algies are expected to provide government, community, researchers, and other stakeholder institutions regarding HAB events in Indonesia and other countries to speed up decision-making in detecting hazard indications and mitigating the effects of HABs.

Compared to an EWS equipped with many physical sensors and apparatuses, Alboom is undoubtedly much cheaper because it uses volunteer humans as sensors. This phenomenon is called social sensing, a paradigm where data are collected from individuals or devices on their behalf, according to *Manik et al. (2019)*. Crowdsourcing and data sharing have been widely applied in various information technology systems, such as geotagging locations on social media, location sharing on various online map platforms and messaging services, and participatory monitoring or reporting systems. However, similar to other community participatory-based applications, the willingness of community members to be actively involved in collecting and sharing the data is critical to the success of implementation.

## Literature review

This subsection presents the theoretical basis, such as the crowdsourcing concept, the technology and acceptance model (TAM), the usability measurement, HABs, Alboom, and similar studies.

### Crowdsourcing

Crowdsourcing terminology still refers to a concept used to outsource a task through collective intelligence in online communities to solve problems, according to *Morschheuser, Hamari & Koivisto (2016)*. However, in subsequent developments, crowdsourcing has become a general term for activities that use the potential intelligence of groups or communities to contribute to problem-solving, knowledge aggregation, content creation, and large-scale data processing. Various needs, contexts, and problems can be applied to crowdsourcing. Several studies have different terms with similar meanings with crowdsourcing for sensing capabilities. For instances, *Ganti, Ye & Lei (2011)* addressed it as crowdsensing, *Kamel Boulos et al. (2011)* named it citizen sensing, and *Liu et al. (2015)* called it social sensing. This sensing is widely applied to data collection for monitoring.

### Technology acceptance model

The technology acceptance model (TAM) was introduced by *Davis (1989)*. The factors that determine whether users accept or reject information technology might vary. However, based on TAM, user acceptance has two critical factors: perceived usefulness and perceived ease of use. Initially, the TAM adopted the theory of reasoned action (TRA) developed

by *Flanders, Fishbein & Ajzen (1975)*. TRA refers to social psychology, which analyses the determinants of conscious behavior, where a person's behavior is determined by the intention to perform the behavior (behavioral intention). Someone tends to use or not use application or information system (IS) because they believe that it will help their work improve. This concept is then interpreted as the perceived usefulness factor. Therefore, perceived usefulness can be construed as a person's level of belief that using an information technology system or application will improve performance. The ease-of-use factor is a person's level of belief that using a system or application will be free from severe effort or free from difficulties. An effort is a limited resource that a person can allocate to perform an activity for which he is responsible. The perceived usefulness and ease of use factor in the TAM has been widely used in IS research.

### Usability

Usability has a broader definition according to ISO 9241:11:2018 (https://www.iso.org/standard/63500.html). Usability is the extent to which specific users can use a system, product, or service to achieve specific goals with effectiveness, efficiency, and satisfaction in particular contexts of use. The definition of effectiveness is the level of accuracy and completeness used by users to achieve specific goals. In comparison, efficiency is the resources used concerning the results achieved. Finally, satisfaction is how users' physical, cognitive and emotional responses result from using systems, products, or services that meet needs and user expectations. A usability evaluation method is an approach to evaluating systems based on human–computer interaction (HCI) concept. This study used the System Usability Scale (SUS) score as part of the usability evaluation. This instrument was initially developed to measure and evaluate products due to the demands and measures product usability at Digital Equipment Co. Ltd. by *Brooke (1996)*. In subsequent development, *Sharfina & Santoso (2016)* adopted the instruments in Indonesian language.

### HABs

A HABs outbreak is typically related to changes in environmental conditions. Some physical and chemical water parameters induce the rapid growth of HAB species. HAB incidents are relatively easy to identify using several indicators of the physical condition of water, such as changes in water color to reddish, brownish, or dark green. Massive algal blooms can form foam, scum, mats, or paint-like features floating on the water's surface. Some HABs are not clearly visible at the water surface in other instances. However, water bodies may appear red, brown, yellow, orange, or dark green. When HABs die off and decompose, they can release unpleasant odors (https://cdc.gov/habs). An increase in water temperature may also help phytoplankton proliferate to form blooms. HAB events also tend to occur with increases in sea surface temperature, which is affected by climate change.

According to *Assmy & Smetacek (2009)*, algae blooms refer to the condition of dense microalgae cells of one or more species due to their unusually rapid growth causing microalgae abundance and biomass increase. On the other hand, HABs refer to the condition when the proliferation of algae species has detrimental effects on humans and other aquatic organisms. Some factors influence the rapid growth of algae biomass that exceeds the average rate. While *Maberly, Van de Waal & Raven (2022)* indicated nutrient

concentration as one of these factors, *Kim et al. (2004)* specified other factors, such as comfortable environmental temperature, salinity, and light intensity. Temperature affects the algae growth rate significantly. Experiments conducted by *Kim et al. (2004)* showed that temperature significantly influences the growth rate of *Cochlodinium polykrikoides*, a causative blooms species that frequently occurs in Lampung Bay. The significant effect of the increased temperature on the algae blooms frequencies was also found by *Yu et al. (2007)*, who analyzed the relationship between temperature increase and HABs appearance in the Northern South China Sea.

### Alboom

Alboom can be a solution for detecting and mitigating HABs. The system's workflow begins with the input of required information, which consists of images of water and the surrounding environment as well as automatic recording of location coordinates and time. Both automatic and manual recording can be used when the user's smartphone device is on the Internet or offline. In offline conditions, the user's data are stored in the device storage and then transmitted to the data server when connected to the Internet. After providing the images, the user (reporter) performs manual qualitative input of environmental conditions related to weather observations, water conditions, and a visual assessment of the situations. Prior to using Alboom for the first time, all users were trained *via* direct and online individual or group training sessions. The training was aimed at standardizing the user's qualitative assessment of the environmental conditions to be inputted into the application. Since all input parameters are in the form of qualitative assessments using simple expressions, as shown by the workflow in Fig. 1, the users had no difficulty in understanding and performing the data input. The data input into the server are then analyzed and verified automatically and relayed to other Alboom users *via* the map viewer. This relay speed is relatively short so that the occurrence of HABs in one place will be immediately known (near real-time) by other users in different places.

Alboom collects data and information on weather and water conditions that could indicate the occurrence of algae bloom on pre-bloom, when the bloom starts to occur, and post-bloom conditions. Pre-bloom alert signs are detected through weather conditions on sunlight intensity and rainfall when the data is recorded and about two to three days before. *Puspasari et al. (2018)* found that algae bloom is triggered by heavy rainfall followed by high sunlight intensity. Therefore, reporting on the rainfall occurrence two or three days before the high intensity of sunlight could become an alert sign for algae bloom and precautions for users to increase their visual monitoring of the color of the water.

Alboom records the water condition data through the users' observation of color changes from the expected condition, water density, the itchy and slimy feel if the water is touched, and the existence of bioluminescence at night to indicate the occurrence of blooming. Color changes in the water are the easiest way to detect the occurrence of algae bloom. The color turns reddish, yellowish, brownish, orange, or dark green, indicating increased algae biomass containing particular pigment colors. For example, *Grate-Lizrraga et al. (2004)* found increased biomass of *Cochlodinium polykrikoides*, which contains chlorophyll-a and c, peridinin, diadinoxantin, and beta-carotene pigments turned the

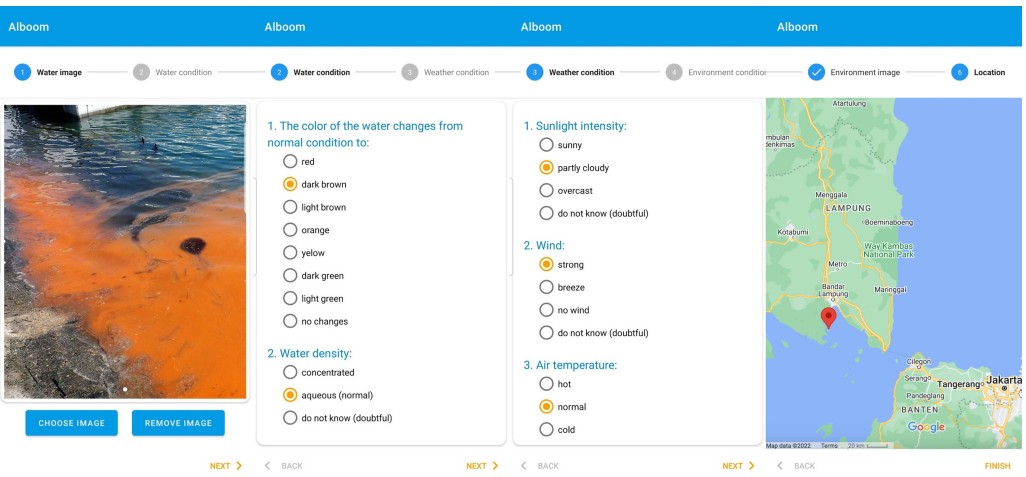

**Figure 1** Uploading Alboom data workflow: capturing images (Photo credit: Ridwan Satria), inputting qualitative assessments of water and weather data, and confirming GPS location (Map data ©2022 Google).

color of the water from normal bluish into reddish. This phenomenon is usually called red tide. Meanwhile, *Gopakumar, Sulochanan & Venkatesan (2009)* showed that the increased biomass of *Noctiluca scintilans* changed the color of the water from normal bluish to dark green. Furthermore, *Puspasari et al. (2018)* found that the sensation of itchy and slimy waters was felt when *Cochlodinium polykrikoides* blooms in Lampung Bay due to the mucus production by the massive concentration of the cells. Another indication of algae blooms is the bioluminescence at night. *Gopakumar, Sulochanan & Venkatesan (2009)* also detected some species of algae, like *Noctiluca scintilans*, contain high fluor concentrations that cause luminescence phenomena in the water when there is limited light. Sometimes, when the increased concentration of algae biomass is not significant but higher than average, users cannot observe the blooms through a visual and touching sensor. However, the occurrence of the algae blooms could still be detected from the post-bloom conditions through the existence of massive dead algae in the form of clumpy and foamy mass floating on the surface water.

The adverse effects of HABs are caused by the drop in oxygen concentration in the water column or the toxic effect of some toxic species, according to *Karlson et al. (2021)*. *Barokah, Putri & Gunawan (2016)* found that 12 species were identified as HABs causative species, and seven of them contain toxic in Lampung Bay. Blooming of toxic algae causes the death of organisms due to the increase of toxic concentration, even though blooming occurs with a low abundance of algae. In this case, the color of the water might not change, and the toxic species blooming could not be indicated visually. Alboom detects the occurrence of toxic species bloom through other data input, such as an itchy and slimy sensation caused by the toxin produced by algae. There is no specific data record on the toxicity conditions in Alboom. However, if the inputted data indicate an occurrence of blooming algae, other
users are notified and then can verify it. In this case, the function of Alboom as sharing system in the community has dominated the role of the early warning system.

### Similar acceptance studies

Similar acceptance studies have been performed when implementing the crowdsourcing concept in various contexts. For example, a study on mobile crowdsourcing technology acceptance in crisis management was conducted by *Yaseen & Al Omoush (2020)* using an extended Unified Theory of Acceptance and Use of Technology (UTAUT). According to their findings, individual and crowd performance expectations, social influence, and perceived risks substantially impact the intention to continue acceptance. Revised UTAUT2 was applied to explain the acceptance of crowdsourcing games by *Wang, Goh & Lim (2020)*. Effort expectancy, hedonic motivation, and social influence directly impacted users' intention to continue playing crowdsourcing games, as well as time-based variations in users' perceptions and acceptance of the games and how their perceptions affected their acceptance. Moreover, *Huang et al. (2020)* used the Push-Pull-Mooring (PPM) theory to understand what factors influenced crowd workers' participation in crowd logistic platforms. Results showed that trust and monetary rewards positively affect crowd workers' motivation to continue working in crowd logistics. Also, another study by *Bakici (2020)* that used an augmented Theory of Planned Behavior (TPB) indicated that attitude and subjective norms significantly impact individuals' intention to participate in crowdsourcing. A summary of previous crowd-based technology acceptance studies is provided in Table 1.

## Research gaps, objectives, and contributions

Nevertheless, acceptance studies for crowd-based technologies using TAM are still rare. Only three relevant studies were found, showing that all proposed TAM hypotheses were accepted. The ease of use and usefulness were essential for stakeholders to use crowdfunding in a study by *Djimesah et al. (2022)*. Perceived ease of use and utility significantly influenced users' intention to use RISCOVID for tracing contacts of persons infected with Covid-19 in a study by *Cruz et al. (2020)*. Moreover, according to a study by *Minkman, Rutten & van der Sanden (2017)*, usefulness, the relevance of the task, and the demonstrability of benefits significantly influenced acceptance of mobile technology for citizen science in water resource management.

### Research gaps

The research gaps to be addressed in this study concern the limited application of the TAM to EWS settings. Although interest in crowdsourcing as a new social computing paradigm is growing, there is a lack of adoption of technology acceptance models to explain the determinants of users' continuous acceptance of crowd-based early warning systems (CBEWS). Table 1 shows studies related to crowdsourcing and its acceptance research in various contexts. However, there is not yet found for EWS contexts, particularly in detecting and mitigating HABs.

**Table 1  Previous crowd-based technology acceptance studies.**

| Ref. | Context | Acceptance model | Results |
|------|---------|------------------|---------|
| *Yaseen & Al Omoush (2020)* | Refugee crisis management | UTAUT | Crowd performance expectancy, the social influence, perceived risks on the individual and crowd levels, and cultural values of collectivism and uncertainty avoidance had a significant influence. However, cultural values of masculinity, power distance, and long-term orientation did not affect the intention. |
| *Wang, Goh & Lim (2020)* | Crowdsourcing games | UTAUT2 | Users' continued intention toward crowdsourcing games was directly influenced by effort expectancy, hedonic motivation, and social influence. Also, time-based variations in users' views and acceptance of the games, as well as how their perceptions affected their acceptance. |
| *Huang et al. (2020)* | Sustainable urban logistics | PPM theory | Monetary rewards and trust had a significant positive impact. However, work enjoyment from previous work and entry barriers for work had a significant negative impact. |
| *Bakici (2020)* | Idea collaboration | TPB | Attitude and subjective norms significantly impacted individuals' intention to participate in crowdsourcing. |
| *Djimesah et al. (2022)* | Crowdfunding in Ghana | TAM | Perceived ease of use and usefulness significantly influenced intention to use. |
| *Cruz et al. (2020)* | Tracing contacts | TAM | Perceived utility (usefulness) as well as ease of use and intention to use had a significant influence on the acceptance of RISCOVID. |
| *Minkman, Rutten & van der Sanden (2017)* | Water resource management | TAM3 | Usefulness, relevance to the task, and the demonstrability of benefits were the important drivers of citizens' behavioral intentions. |

## Objectives

The objectives of this study were to investigate and examine factors that determine users' acceptance of CBEWS by extending the original version of TAM and incorporating other variables. This study used the TAM because it considers users' technical experiences and beliefs about how technology might influence their behavior in a crowd-based early warning ecosystem. The TAM was a powerful and robust prediction model for understanding user adoption of technology in many circumstances, according to a meta-analysis by *King & He (2006)* of 88 studies in diverse domains. The original TAM was created to describe end users' readiness to use new technology in businesses. Also, the SUS score was used in this study's context to determine whether the usability measure affects individuals' intention to accept and use CBEWS long-term. Therefore, this study proposed a new model to reveal the determinants and fill the research gap for this specific context.

## Contributions

The contributions of this study are twofold. First, this study investigated factors that affect the acceptance and use of CBEWS using TAM theory, an IS-based approach. The fundamental determinants used in the original TAM model need to elaborate on what factors need to be concerned in developing and implementing CBEWS. In addition to native TAM variables, this study also incorporated other factors by using variables that had significant effects based on the findings of previous studies, such as awareness, rewards, and

 

social influence. Second, this study investigated whether the application usability measure using the SUS score, an HCI-based approach, influences the continual use intention. In general, theory-driven research like this study promotes a better understanding of the attitudes and behaviors influencing a particular action. For example, organizations or experts can build applicable methods to advertise positive responses by understanding what motivates users to use CBEWS on a daily basis. To our knowledge, this study is the first investigation of CBEWS usability and acceptance analysis, particularly in detecting and mitigating HAB events. The results of this study may also be applied in other CBEWS use cases.

## RESEARCH MODEL AND HYPOTHESIS

This study developed a model that supports the intention to use CBEWS continuously. Specifically, this study proposed that usability (USA) could influence the perceived ease of use (PEU), and the later could affect perceived usefulness (PUF). Then, perceived ease of use (PEU), perceived usefulness (PUF), and awareness (AWA) could influence attitudes (ATT) toward using CBEWS. Meanwhile, social influence (SOC), attitude (ATT), and reward (REW) could significantly affect the continual use intention (INT). Figure 2 shows the proposed model.

This study considered usability measures as a factor in the research model. In this sense, a website's usability definition by *Choros & Muskala (2009)* was adopted. Usability is defined as a set of layout, structure, arrangements, typography, and many other aspects that make an application simple and easy to use. The SUS score was specifically used to measure usability in this study. Furthermore, a comprehensive study by *Tao et al. (2020)* integrated usability, in particular usability testing performed by users to accomplish particular tasks, and TAM to understand young consumers' adoption of a health information portal. As a result, subjective usability influenced perceived ease of use positively. Moreover, *Mlekus et al. (2020)* also combined usability using a user experience (UX) questionnaire with TAM. The results showed that usability, particularly perspicuity and dependability, significantly affected the perceived ease of software R (https://www.r-project.org). Following these successful works, usability was considered a factor in the research model. However, unlike previous studies, this study used the SUS score to assess UX characteristics. This study hypothesized that the usability of CBEWS could positively affect perceived ease of use.

**H 1** *Usability of CBEWS positively affects perceived ease of use.*

According to TAM, one of its native variables, perceived ease of use, influences the other native variable, perceived usefulness. A previous study of Ghanaian crowdfunding by *Djimesah et al. (2022)* proved this relationship. It indicates that perceived ease of use plays a critical positive factor affecting users' acceptance of participatory-based technologies or systems. Furthermore, in the acceptance study of Covid-19 by *Akther & Nur (2022)*, the perceived ease of use positively affected the attitude toward behavior. Therefore, based on prior studies, this native TAM variable was used as a factor in the proposed model. This factor represents the users' opinion regarding the ease of using the Alboom application in

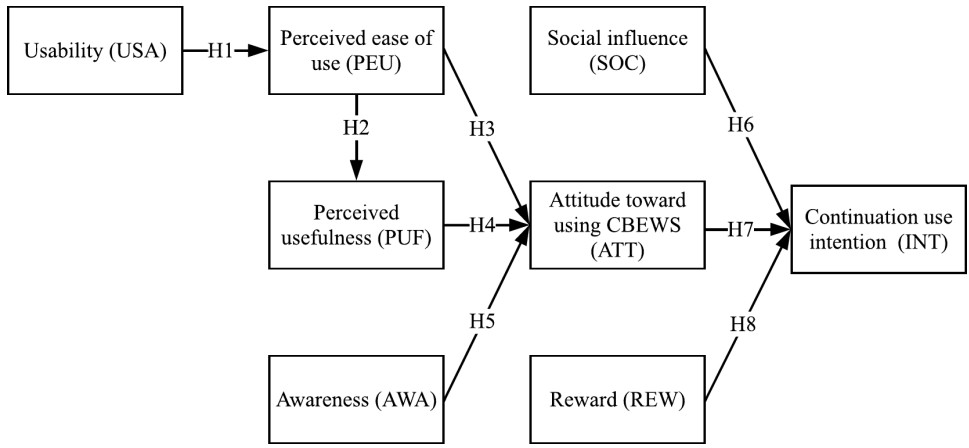

**Figure 2   Proposed acceptance model of crowd-based early warning systems for harmful algal blooms.**

this study. In particular, perceived ease of use was expected to positively affect the perceived usefulness and the users' attitude to continue using CBEWS.

**H 2**  *Perceived ease of use positively influences perceived usefulness.*

**H 3**  *Perceived ease of use positively influences attitudes towards using CBEWS.*

Another factor in the proposed model was perceived usefulness. According to the acceptance study of e-procurement by *Brandon-Jones & Kauppi (2018)*, perceived usefulness positively affected users in accepting the technology. Furthermore, as again shown by *Akther & Nur (2022)*, the perceived usefulness influenced the attitude toward Covid-19 acceptance. This study used this native TAM variable to indicate that using Alboom could benefit its users, particularly their job performance. Therefore, perceived usefulness was believed to impact users' attitudes toward using CBEWS positively.

**H 4**  *Perceived usefulness positively influences attitudes towards using CBEWS.*

As the area with the most frequent HAB events, Lampung Bay has experienced HAB events since 2004. Therefore, such frequent events may facilitate Lampung Bay's coastal communities to better understand and know about HABs. However, *Aditya et al. (2015)* reported that less than 48.6% of respondents in their study in the Lampung coastal area knew the indicators of HAB occurrence. Similarly, *Hidayati (2020)* reported that only up to 50% of Lampung Bay coastal communities know that HABs could last for several days and cause fish death. These studies indicate that many of the Lampung Bay coastal communities are still unaware of HABs and their direct negative impacts on their economy and public health in general. Few to no HAB cases were reported from other areas in Indonesia, primarily due to the lack of HAB awareness in the coastal community and the absence of HAB reporting or early warning systems. Recent studies by *Akther & Nur (2022)*, *Rahman & Sloan (2017)* and *Mashal, Shuhaiber & Daoud (2020)* showed that people's awareness is a significant factor in accepting COVID-19 vaccination, mobile commerce, and smart homes, respectively. Thus, an awareness factor was included in the proposed model.

It is the magnitude of knowledge users possess about the potential dangers of HABs. Understanding the risks and hazards were believed to affect the users' attitude toward using Alboom positively. Therefore, it was expected that awareness could positively affect users' attitudes toward using CBEWS.

**H 5**  *Awareness positively influences attitudes towards using CBEWS.*

Social influence can be explained as a factor in which users are affected by other people (*e.g.*, families, friends, and neighborhoods) to use a system or to be involved in an activity. Previous crowdsourcing studies by *Yaseen & Al Omoush (2020)* and *Wang, Goh & Lim (2020)* showed that social influence is an essential factor. In particular, *Mashal, Shuhaiber & Daoud (2020)* explained that social influence had significant positive impacts on people's intention to use smart home applications (*e.g.*, smart TV, smart fridge, and smart lights). Meanwhile, *Panopoulou, Tambouris & Tarabanis (2021)* stated that social influence had significant positive effects on people's intention to use an e-participation system, Puzzled by Policy (PbP). Based on the findings of those prior studies, social influence was included in the research model. Specifically, it was expected that social influence could positively affect users to continue using CBEWS.

**H 6**  *Social influence positively affects continuation use intention of CBEWS.*

Attitude was a critical factor in accepting a system based on a crowdsourcing study by *Bakici (2020)*. Moreover, *Brandon-Jones & Kauppi (2018)* claimed that attitude toward a system had a significant positive impact on users to continue using the e-procurement. Therefore, attitude was used as a factor in the proposed model. Specifically, the attitude toward using CBEWS was expected to affect users positively to the continuation use intention.

**H 7**  *Attitude toward using CBEWS positively affects continuation use intention.*

This study presumed that obtaining a reward could be one reason users use Alboom continuously. This presumption was based on previous studies by *Cappa, Rosso & Hayes (2019)*, *Huang et al. (2020)* and *Ye & Kankanhalli (2017)* that stated reward had a significant positive effect on increasing the number of users' participation and influencing the users to participate continuously in a crowdsourcing environment. Because Alboom relies on a crowdsourcing approach, it was believed that reward might positively affect users' intention to use Alboom continuously. Thus, the reward factor was proposed in the research model. In particular, it was expected that reward could positively affect users to continue using CBEWS.

**H 8**  *Reward positively affects continuation use intention of CBEWS.*

## METHODS

This study used quantitative methods, and respondents were asked to state their agreement with certain statements using a Likert scale of 1 to 5, where 1 indicated "strongly disagree", 2 indicated "disagree", 3 indicated "neutral", 4 indicated "agree", and 5 indicated "strongly agree". The survey was approved by the Research Ethics Committee on Social Studies and Humanities, National Research and Innovation Agency with approval number of
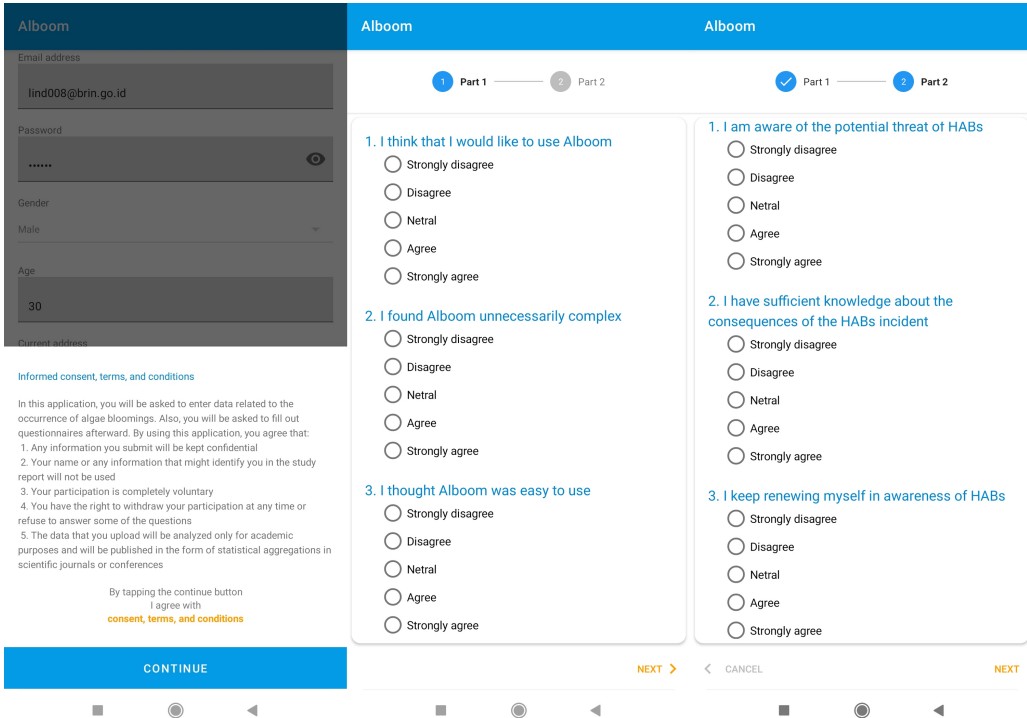

**Figure 3** Informed consent and survey questionnaires of Alboom mobile application.

164/KE.01/SK/8/2022. This study also complied with all relevant ethical regulations, and informed consent was obtained from all participants.

## Data collection

Data collection was performed using online survey. Before collecting questionnaire data, two general lecture and training workshop webinars (http://alboom.mict.id) were conducted to increase participants' awareness about the role of technology and community participation in detecting and mitigating HAB incidence. Participants were part of the EWS stakeholders for HABs consisting of fish farmers, fishers, governments, fishery instructors, researchers, and students.

In the first webinar, Alboom was introduced to the public for the first time. Then, the attendees were asked to install the mobile application on their smartphone devices. The users must input their profile data when registering themselves in the application. Informed consent was obtained before creating the user account. The new users were required to agree to the terms and conditions, as shown in Fig. 3, such as:

- Any information users submit will be kept confidential.
- Users' name or any information that might identify their profile in the study report will not be used.
- Users' participation is completely voluntary.
- Users have the right to withdraw their participation at any time, or refuse to answer some of the questions.

**Table 2  Usability instruments.**

| Code | Questionnaire |
| --- | --- |
| sus1 | I think that I would like to use Alboom |
| sus2 | I found Alboom unnecessarily complex |
| sus3 | I thought Alboom was easy to use |
| sus4 | I think that I would need the support of a technical person to use Alboom |
| sus5 | I found the various functions in Alboom were well integrated |
| sus6 | I thought there was markedly well inconsistency in Alboom |
| sus7 | I would imagine that most people would learn to use Alboom very quickly |
| sus8 | I found Alboom very cumbersome to use |
| sus9 | I felt very confident using Alboom |
| sus10 | I required to learn a lot of things before I could get going with Alboom |

- The data that users upload will be analyzed only for academic purposes and will be published in the form of statistical aggregations in scientific journals or conferences.

After consenting and finishing registration, users were asked to collect data using Alboom regularly. Users who consistently contributed data six times between the first and the second webinar have been rewarded a certificate of appreciation. After the second webinar, the questionnaire was distributed, and all users were invited to respond through the mobile application.

### Research instrument

The research instrument was divided into two parts. The first part consisted of standard SUS questionnaires, as shown in Table 2. The instrument has 10 questions, where odd items are positive statements, while even ones are negative ones. Meanwhile, the second part of the questionnaire consisted of the augmented TAM instruments is shown in Table 3.

### Analysis method

The analysis method was performed using statistics tools available in SmartPLS 4 (https://smartpls.com/) to calculate all statistical computations. The model was analyzed using partial least squares structural equation modeling (PLS-SEM) algorithm. Before performing analyses, reliability and validity tests were conducted by measuring Cronbach's alpha, factor loading, average variance extracted (AVE), and discriminant validity. Meanwhile, calculating SUS score simple, only requiring odd and even question numbers to be distinguished. If the number is odd, then the result of the respondent's value is reduced by 1; if the number is even, then the value is five minus the value of the respondent. The scores of the ten questions are summed and then multiplied by 2.5 as shown by Eq. (1)

**Table 3  Acceptance instruments.**

| Construct | Items | Questionnaires | Adopted from |
|---|---|---|---|
| PEU | peu1 | Learning to use Alboom is easy for me | *Djimesah et al. (2022); Akther & Nur (2022)* |
| | peu2 | It is easy for me to become proficient in using Alboom | |
| | peu3 | The use of Alboom is clear and easy to understand | |
| | peu4 | Overall, it is easy for me to use Alboom | |
| PUF | puf1 | Alboom provides useful information to me | *Brandon-Jones & Kauppi (2018); Akther & Nur (2022)* |
| | puf2 | Alboom adds to my knowledge about HABs prevention efforts | |
| | puf3 | Using Alboom is relevant or useful for my work | |
| AWA | awa1 | I am aware of the potential threat of HABs | *Rahman & Sloan (2017); Mashal, Shuhaiber & Daoud (2020); Akther & Nur (2022); Manik et al. (2022)* |
| | awa2 | I have sufficient knowledge about the consequences of the HABs incident | |
| | awa3 | I keep renewing myself in awareness of HABs | |
| | awa4 | I share HABs knowledge with my colleagues to increase awareness | |
| SOC | soc1 | I need to use Alboom according to my colleagues' opinions | *Yaseen & Al Omoush (2020); Panopoulou, Tambouris & Tarabanis (2021)* |
| | soc2 | According to people who influence my behavior, I must use Alboom | |
| | soc3 | If coastal communities feel helped by Alboom, then I must use this application | |
| ATT | att1 | I think using Alboom is a good idea | *Akther & Nur (2022), Brandon-Jones & Kauppi (2018)* and *Wilis & Manik (2022)* |
| | att2 | I have a positive attitude towards using Alboom | |
| | att3 | All things considered, the use of Alboom is recommended | |
| | att4 | I think using Alboom is interesting and fun | |
| REW | rew1 | I hope to receive a reward (e.g., a certificate, credit or otherwise) for my contribution to Alboom | *Cappa, Rosso & Hayes (2019), Huang et al. (2020)* and *Ye & Kankanhalli (2017)* |
| | rew2 | The more rewards I get, the more I want to contribute to Alboom | |
| | rew3 | I am satisfied with the rewards given in using Alboom | |
| INT | int1 | Based on my experience, I will most likely continue to contribute to Alboom | *Djimesah et al. (2022), Bakici (2020)* and *Cruz et al. (2020)* |
| | int2 | I will recommend others to use Alboom | |
| | int3 | I plan to use Alboom often in the future | |

where $U_i$ refers to the rating of the $i$th item.

$$SUSscore = 2.5 \times \left[ \sum_{n=1}^{5} (U_{2n-1} - 1) + (5 - U_{2n}) \right]. \tag{1}$$

## RESULTS

This section presents the collected data and quantitative analysis results. Both analyses on SUS and TAM are explained using statistical measurements. The dataset obtained was uploaded to a public repository (https://doi.org/10.5281/zenodo.7491272).

**Table 4 Demographics of the participants.**

| Characteristics (Respondents = 104) | | Percentage |
|---|---|---|
| **Gender** | Male | 49% |
| | Female | 51% |
| **Age** | 17–25 | 54% |
| | 26–34 | 14% |
| | 35–43 | 10% |
| | 44–52 | 16% |
| | 53–61 | 6% |
| **Education** | Doctoral degree | 9% |
| | Master degree | 21% |
| | Bachelor degree | 15% |
| | High school | 55% |
| **Profession** | College student | 54% |
| | Researcher | 20% |
| | University lecturer | 9% |
| | Fishery instructor | 8% |
| | Fisher | 5% |
| | Teacher | 2% |
| | Government employee | 2% |
| **Upload count** | 0 times | 32% |
| | 1–5 times | 26% |
| | 6 times | 27% |
| | More than 6 times | 15% |

## Participant demographics

The first webinar was attended by 488 people. Meanwhile, only 138 participants installed Alboom (https://appho.st/d/JM80Ljzf). Alboom was introduced continuously to the stakeholders between the first and the second webinars, and afterward. Thus, the number of new users increased to 223. However, only 109 people have ever uploaded data using Alboom at least once. All users were offered to respond to the questionnaires, and 104 of 223 people provided responses.

Table 4 shows the demographic information of the respondents. The proportion of men and women in this survey was balanced. Most respondents were 17–25 years old (54%), which indicates that the majority were digital natives. Based on reported job distributions and education levels, most of the respondents were well educated. Also, 32% of respondents have installed and used Alboom but never uploaded data. Meanwhile, 42% of respondents reserved the reward of certification because they uploaded data using Alboom six times or more.

## System usability scale (SUS) measurement

In order to determine whether the SUS instruments had good reliability, it is necessary to calculate Cronbach's alpha value, which is a test score reliability coefficient to measure how closely related a set of items are as a group. The results of the calculation, as shown

**Table 5  SUS score and reliability of measurements.**

| No | Code | Items value | | | | | $\sum$ Norm | % two highest values | $\alpha$ |
|----|------|------|------|------|------|------|------|------|------|
| | | 1 | 2 | 3 | 4 | 5 | | | |
| 1 | sus1 | 1% | 0% | 18% | 55% | 26% | 3.05 | 81% | 0.81 |
| 2 | sus2 | 11% | 64% | 22% | 1% | 2% | 2.81 | 75% | 0.79 |
| 3 | sus3 | 1% | 0% | 15% | 57% | 27% | 3.09 | 84% | 0.79 |
| 4 | sus4 | 10% | 50% | 19% | 15% | 6% | 2.42 | 60% | 0.81 |
| 5 | sus5 | 0% | 7% | 21% | 59% | 13% | 2.79 | 72% | 0.78 |
| 6 | sus6 | 4% | 46% | 43% | 7% | 0% | 2.47 | 50% | 0.78 |
| 7 | sus7 | 0% | 6% | 16% | 54% | 24% | 2.96 | 78% | 0.78 |
| 8 | sus8 | 10% | 70% | 15% | 5% | 0% | 2.85 | 80% | 0.77 |
| 9 | sus9 | 0% | 14% | 14% | 55% | 16% | 2.73 | 71% | 0.77 |
| 10 | sus10 | 1% | 21% | 14% | 51% | 13% | 1.47 | 22% | 0.81 |
| | SUS score | 66.59 | | | | | | | |

in the Cronbach's alpha ($\alpha$) column in Table 5, have values above 0.7. Because reliability theory, according to *Nunnally (1975)*, requires a Cronbachs's alpha value of at least 0.7, the reliability of the variables and the level of internal consistency of the instrument are confirmed.

The SUS instruments, as shown in Table 2, have ten questions that contain both positive and negative meanings. For example, questions on numbers 1, 3, 5, 7, and 9 have positive connotations, while questions 2, 4, 6, 8, and 10 have negative meanings. This difference in positive and negative statements will result in a different grade. If a question is positive, selecting a higher value (*e.g.*, strongly agree) will yield a true value. In contrast, if the question is negative, a lower value (*e.g.*, strongly disagree) yields a higher score. Therefore, the values must be normalized to find the absolute highest value.

The normalized results shown in Table 5 were calculated by reducing the value input from the respondent by lifting one (1) for positive questions. In contrast, by reducing five (5) by the value of the respondent, the value will be between zero (0) and four (4). Table 5 also shows the percentage of respondent values one (1) to five (5). To calculate the optimal ratio, in the % two highest values column, the calculation adds one (1) and two (2) if the question is negative, and if the question is positive, it adds three (3) and four (4). The tenth item has an average normalization result below 1.47 with 22%. This item stated that users had to learn many things before they could use Alboom, which indicates that the respondents required adaptation to use Alboom.

Referring to Eq. (1), the normalized results were then multiplied by 2.5 to determine the level of usability perception in the Alboom application. The average score of all respondents, as calculated using Eq. (2), is the final SUS score, which ranges from 0 to 100, given the number of respondents, *n*, 104. Table 5 shows the SUS score for the Alboom application of 66.59. This value is sufficient, has a grade "D" scale, adjective ratings of "OK", and a

high-marginal acceptability range, according to *Bangor, Kortum & Miller (2009)*.

$$\overline{SUSscore} = \frac{1}{n}\sum_{i=1}^{n} SUSscore_i. \tag{2}$$

## Assessment of measurement model

The measurement model was assessed based on factor loading, construct reliability using Cronbach's alpha, AVE parameter, and discriminant validity. As shown in Table 6, seven construction item indicators (AWA, PEU, PUF, SOC, ATT, REW, and INT) had loading values between 0.56−0.96. A factor loading is the correlation coefficient for the variable and factor. It describes the variance the variable explains on that particular factor. According to *Hair Jr et al. (2014)*, the ideal allowable factor loading should exceed 0.7, which indicated that the factor removed sufficient variance from the variable. Thus, the construct indicators of awa1, awa2, att1, and soc3 were dropped because their loading values were below 0.7. Furthermore, all Cronbach's alpha values were more significant than 0.7, indicating that all constructs were reliable.

Also, convergent validity was assessed using AVE. As shown in Table 6, all AVE values in the six construct parameters exceeded the minimum threshold value of 0.5, according to *Fornell & Larcker (1981)*, which indicated that the variance captured by the construct was larger than the variance due to measurement error. Thus, all constructs were valid. Table 7 shows the discriminant validity of each indicator for all construct parameters. According to *Monecke & Leisch (2012)*, the discriminant validity values of less than 0.2 are not shown in the output. The largest values for each indicator were in the construct parameter, and the built indicators were appropriate for measuring the construct parameters. Based on these results, the measurement model was satisfactory.

## Assessment of structural model

Bootstrapping testing was performed to test the significance of the effect of one variable on another. This study accepted a hypothesis if the $p$-value is less than a significant level of 0.05. Therefore, all hypotheses in the proposed model were supported except H8, as shown in Table 8.

As expected, PEU was found to be significantly affected by USA (H1: $\beta = 0.72$; $p < 0.001$). Furthermore, PEU significantly affected PUF (H2: $\beta = 0.55$; $p < 0.001$) but PEU had small significant effect on ATT (H3: $\beta = 0.22$; $p = 0.046$). PUF and AWA positively predicted ATT (H4: $\beta = 0.36$; $p < 0.001$, H5: $\beta = 0.30$; $p = 0.001$). Moreover, INT was found to be significantly influenced by ATT and SOC. Based on the magnitude of the path coefficient value, which is significant in each construct, the attitude toward using CBEWS plays the most important role in determining a person's desire to use CBEWS subsequently (H7: $\beta = 0.59$; $p < 0.001$). The path coefficient value was more than twice the coefficient of the path from social influence to intention (H6: $\beta = 0.25$; $p = 0.001$). Nevertheless, REW did not significantly positively predict INT (H8: $\beta = -0.04$; $p = 0.638$).

**Table 6  Measurement model.**

| Construct | Items | Loading | Cronbach's α | AVE |
|---|---|---|---|---|
| **USA** | SUS score | 1.00 | 1.00 | 1.00 |
| **PEU** | peu1 | 0.91 | 0.90 | 0.77 |
| | peu2 | 0.86 | | |
| | peu3 | 0.85 | | |
| | peu4 | 0.89 | | |
| **PUF** | puf1 | 0.91 | 0.84 | 0.76 |
| | puf2 | 0.93 | | |
| | puf3 | 0.77 | | |
| **AWA** | awa1 | 0.67[a] | 0.82 | 0.85 |
| | awa2 | 0.69[a] | | |
| | awa3 | 0.87 | | |
| | awa4 | 0.84 | | |
| **SOC** | soc1 | 0.89 | 0.87 | 0.88 |
| | soc2 | 0.90 | | |
| | soc3 | 0.69[a] | | |
| **ATT** | att1 | 0.56[a] | 0.91 | 0.84 |
| | att2 | 0.91 | | |
| | att3 | 0.92 | | |
| | att4 | 0.90 | | |
| **REW** | rew1 | 0.94 | 0.95 | 0.91 |
| | rew2 | 0.97 | | |
| | rew3 | 0.96 | | |
| **INT** | int1 | 0.90 | 0.85 | 0.77 |
| | int2 | 0.83 | | |
| | int3 | 0.90 | | |

[a] Drop items

## DISCUSSION

Based on the results of this study, the attitude was the most influential factor that affects the continual intention to use. This finding supports TAM studies by *Akther & Nur (2022)*; *Brandon-Jones & Kauppi (2018)*, and implies that latent variables significantly influence attitude should be identified. This study also found that perceived usefulness, as in a TAM study by *Akther & Nur (2022)*, positively affected attitude with the highest effect. While typical TAM studies, like in *Akther & Nur (2022)*, found that perceived ease of use strongly predicted users' attitudes, results suggested that this strong association was not always present. In fact, a study by *Brandon-Jones & Kauppi (2018)* showed that perceived ease of use had no significant effect on users' attitudes. Although the perceived ease of use had a small effect on the attitude in this study, it influenced the usefulness significantly, as supported by studies in *Cruz et al. (2020)* and *Panopoulou, Tambouris & Tarabanis (2021)*. On the other hand, the usability affected perceived ease of use positively. This finding corroborates studies by *Mlekus et al. (2020)* and *Tao et al. (2020)* even though these studies have different approaches to assessing usability. Therefore, it implies that the higher
**Table 7  Discriminant validity check.**

|  | USA | PEU | PUF | AWA | SOC | ATT | REW | INT |
|---|---|---|---|---|---|---|---|---|
| *SUS score* | 1.00 | . | . | . | . | . | . | . |
| *peu1* | . | 0.91 | . | . | . | . | . | . |
| *peu2* | . | 0.85 | . | . | . | . | . | . |
| *peu3* | . | 0.85 | . | . | . | . | . | . |
| *peu4* | . | 0.89 | . | . | . | . | . | . |
| *puf1* | . | . | 0.91 | . | . | . | . | . |
| *puf2* | . | . | 0.93 | . | . | . | . | . |
| *puf3* | . | . | 0.77 | . | . | . | . | . |
| *awa3* | . | . | . | 0.92 | . | . | . | . |
| *awa4* | . | . | . | 0.92 | . | . | . | . |
| *soc1* | . | . | . | . | 0.94 | . | . | . |
| *soc2* | . | . | . | . | 0.94 | . | . | . |
| *att2* | . | . | . | . | . | 0.90 | . | . |
| *att3* | . | . | . | . | . | 0.93 | . | . |
| *att4* | . | . | . | . | . | 0.91 | . | . |
| *rew1* | . | . | . | . | . | . | 0.94 | . |
| *rew2* | . | . | . | . | . | . | 0.97 | . |
| *rew3* | . | . | . | . | . | . | 0.96 | . |
| *int1* | . | . | . | . | . | . | . | 0.90 |
| *int2* | . | . | . | . | . | . | . | 0.83 |
| *int3* | . | . | . | . | . | . | . | 0.90 |

**Table 8  Structural model hypothesis.**

| Hypothesis | Path | Path coeff. ($\beta$) | $p$-value | Supported |
|---|---|---|---|---|
| H1 | USA → PEU | 0.72 | 0.000 | Yes |
| H2 | PEU → PUF | 0.55 | 0.000 | Yes |
| H3 | PEU → ATT | 0.22 | 0.046 | Yes |
| H4 | PUF → ATT | 0.36 | 0.000 | Yes |
| H5 | AWA → ATT | 0.30 | 0.001 | Yes |
| H6 | SOC → INT | 0.25 | 0.001 | Yes |
| H7 | ATT → INT | 0.59 | 0.000 | Yes |
| H8 | REW → INT | −0.04 | 0.638 | No |

usability of a system, which is determined by how well its features suit users' needs and contexts, would lead to a higher perception of ease of use. Furthermore, the more users perceive a system as easy to use, the more users perceive the system as helpful in achieving the users' goals.

Moreover, awareness factor, which refers to knowledge of indications and impacts of hazards that an EWS solves, introduced in this study significantly influenced the attitude toward continuation use intention. This finding supports (*Rahman & Sloan, 2017*; *Mashal, Shuhaiber & Daoud, 2020*; *Akther & Nur, 2022*), although these prior studies had different contexts. However, unlike many other crowdsourcing studies, reward did not significantly

affect continuation use intention in this research. The context of this study related to an EWS could be one possible reason. Suppose the users know that the potential dangers would impact them or others. In that case, they intend to use the application without rewards because it would be useful to detect and mitigate the hazards. Furthermore, social influence had a more significant effect on use intention. This finding agrees with *Yaseen & Al Omoush (2020)*, *Wang, Goh & Lim (2020)* and *Panopoulou, Tambouris & Tarabanis (2021)*.

## Research implications

Although many studies debate whether an HCI-based approach, like usability, could be combined with an IS-based approach, this study showed a successful integration of usability, particularly SUS score, into TAM. Nevertheless, studies by *Pal & Vanijja (2020)* and *Albastaki (2022)* showed that perceived ease of use had high similarity with usability, particularly SUS score. Therefore, as an implication of the research, usability could be optionally included in the future acceptance models if perceived ease of use or a similar variable is already incorporated. This study showed a strong relationship between the two variables, where usability significantly influenced perceived ease of use.

It is worth revisiting the reward factor in subsequent research. Most crowd-based technology acceptance studies found reward as a significant driver of use. It could be because those studies used monetary rewards. Meanwhile, in this study, a general term of reward was used. In fact, non-financial rewards, such as certificates, credits, et cetera, were given to participants. However, due to a limited budget and regulations, financial rewards were not provided. Therefore, further studies are needed to investigate the monetary reward and its relationship with intention to use, attitude, or even perceived usefulness.

## Practice implications

Even though perceived ease of use and usability is considered a similar variable, in practice, it is suggested that organizations or practitioners conduct both analysis, IS-based approaches first, followed by HCI-based approaches. For example, suppose perceived ease of use or another similar factor is found to be significant. In that case, the degree to which users can use the application to achieve quantified objectives with efficiency, satisfaction, and effectiveness in a quantified context of use should be measured. Then, if the usability score is below average, the application should be improved. In this study case, the usability of Alboom should be revamped because the SUS score was only 66.59.

Reward was not a determinant of the continual use intention of CBEWS. Therefore, this result might imply that providing a reward is not a solution for organizations or practitioners to boost application use. A reward might not guarantee that users will continue to use CBEWS in the future. However, awareness positively influenced attitude toward using CBEWS. Based on this finding, it is suggested that organizations or practitioners should frequently increase citizens' awareness regarding the hazards' context. In this study case, webinars about HABs were organized for citizens to educate them regarding the indications and impacts of HABs. Also, a live on-field training could be conducted to demonstrate how the entire system works. Furthermore, social influence positively

affected the continual use intention, which might indicate that the more users that use CBEWS, the more likely that other users are to be socially influenced to also use CBEWS. Therefore, organizations should encourage inspired people, such as managers and leaders or respectable persons, to embrace CBEWS and persuade others to use it on a long-term basis.

## Limitations and threats to validity

This subsection identifies limitations and threats to the validity of this research and discusses how they can possibly be addressed. This study considers four validity threats: internal, external, construct, and conclusion validity.

To control for the internal validity threat of multiple submissions from the same participant, users were asked to log in to the mobile application before submitting the response to the questionnaire. Thus, it was ensured that participants who completed the questionnaires had installed and used Alboom. All users were encouraged to respond to the questionnaires. Reminders were sent to the users' mobile applications and emails every day.

Although respondents consisted of 47% of the population of Alboom's users, this does not mean the results can be generalized. The respondents were dominated by students (54%), researchers (20%), and university lecturers (9%), which indicates that current Alboom users are primarily scholars. The primary target users of Alboom in the future will be fish farmers, fishermen, and others that have primary related jobs in coastal waters because they spend most of their time in the field, which makes them available at any time to upload data. However, respondents from the most expected users, such as fishery instructors (8%) and fishers (5%), were limited in this study. Also, fishers currently using Alboom are not purely voluntary because they were facilitated with smartphone devices financed by this project budget. The variable in this study, the awareness factor, had a significant influence on the attitude toward using CBEWS and might be affected by the background of the highly educated respondents, which leads to a stronger understanding. These limitations are potential threats to external validity. However, in future work, this study will be repeated when the number of Alboom non-scholar users increases.

Threats to construct validity were manageable because Cronbach's alpha and factor loading for each question in SUS and TAM questionnaires were beyond the standard value of 0.7. If their value was less than 0.7, the question items were dropped. These items were primarily adapted from highly cited studies on TAM and SUS. Furthermore, convergent validity was checked using AVE measures, and discriminant validity was also tested to ensure that the constructs that should have no relationship indeed do not have any relationships. Only reliable and valid items were considered in the SEM analysis.

For statistical conclusion validity, the SEM technique was used in this study to fit a theoretical model to the data. Model fit indicators indicate that the model is sound. SEM further improves conclusion validity by adjusting for multiple comparisons, measurement error (by inferring latent variables from observable variables), and testing the full model (rather than one hypothesis at a time). Alternative path modeling techniques, such as

partial least squares path modeling like Bayesian approaches, are regarded as inferior to SEM, according to *Rnkk & Evermann (2013)*.

**Future works**

More recent theories should be implemented in the future. For example, UTAUT or its second version should be used to investigate other factors that could positively influence attitude, such as facilitating conditions, performance, and effort expectancy, because the attitude toward using CBEWS was the most influential factor in this research. Moreover, latent variables could be added in subsequent studies, based on the second version of the TAM created by *Venkatesh & Davis (2000)* or the third version by *Venkatesh & Bala (2008)*. Also, the actual use variable could be examined in the future by looking at actual usage data. Specific actions could be implemented to achieve particular targets by knowing specific factors to promote positive behavior.

Furthermore, only one CBEWS was examined in this study. Future works should investigate whether the same results are acquired in other CBEWSs. Also, usability addressed in this study was only measured with a single approach. Future research could check whether other usability measures, survey-based or even usability testing approaches, would generate the same results. Moreover, individuals' perceptions of technology may evolve over time. As a result, the current findings could serve as a starting point for future longitudinal studies into the shifting roles of predictors in users' acceptance and subsequent use of CBEWS.

# CONCLUSIONS

A usability and acceptance analysis of CBEWS was conducted in this study. The research model designed in this study enriched the understanding of CBEWS, particularly in detecting HAB incidents and mitigating their effects. This study's findings strongly indicate that improving the knowledge and awareness of a local coastal community about HABs and their potential negative impacts *via* education will be more effective than providing rewards to users. In addition, formal social influence on the human resources of government and non-government institutions, particularly those working or living near high-risk areas, also offers alternative support in increasing the usage of CBEWS and other similar crowdsensing applications to prevent and mitigate the potential dangers.

# ACKNOWLEDGEMENTS

The authors would like to thank all the Knowledge Engineering Research Group members for their suggestions and valuable feedback about this study. In addition, the other members of the SATREPS Mariculture Project from Japan and Indonesia are also thanked for their indirect contributions to the study.

### Funding

This work was supported by the Japan International Cooperation Agency, Japan Science Technology Agency, and the Indonesian Ministry of Marine Affairs and Fisheries through the Science and Technology Research Partnership for Sustainable Development (SATREPS) Mariculture Project Grant "Optimizing Mariculture Based on Big Data with Decision Support System." The funders had no role in study design, data collection and analysis, decision to publish, or preparation of the manuscript.

### Grant Disclosures

The following grant information was disclosed by the authors:
The Japan International Cooperation Agency.
Japan Science Technology Agency.
the Indonesian Ministry of Marine Affairs.
Fisheries through the Science and Technology Research Partnership for Sustainable Development (SATREPS) Mariculture Project.

### Competing Interests

The authors declare there are no competing interests.

### Author Contributions

- Lindung Parningotan Manik conceived and designed the experiments, performed the experiments, analyzed the data, prepared figures and/or tables, authored or reviewed drafts of the article, and approved the final draft.
- Hatim Albasri conceived and designed the experiments, performed the experiments, authored or reviewed drafts of the article, and approved the final draft.
- Reny Puspasari conceived and designed the experiments, performed the experiments, authored or reviewed drafts of the article, and approved the final draft.
- Aris Yaman conceived and designed the experiments, performed the experiments, analyzed the data, prepared figures and/or tables, authored or reviewed drafts of the article, and approved the final draft.
- Shidiq Al Hakim conceived and designed the experiments, performed the experiments, analyzed the data, prepared figures and/or tables, authored or reviewed drafts of the article, and approved the final draft.
- Al Hafiz Akbar Maulana Siagian conceived and designed the experiments, performed the experiments, analyzed the data, prepared figures and/or tables, authored or reviewed drafts of the article, and approved the final draft.
- Siti Kania Kushadiani conceived and designed the experiments, performed the experiments, analyzed the data, prepared figures and/or tables, authored or reviewed drafts of the article, and approved the final draft.
- Slamet Riyanto conceived and designed the experiments, performed the experiments, analyzed the data, prepared figures and/or tables, authored or reviewed drafts of the article, and approved the final draft.

- Foni Agus Setiawan performed the experiments, analyzed the data, authored or reviewed drafts of the article, and approved the final draft.
- Lolita Thesiana performed the experiments, prepared figures and/or tables, and approved the final draft.
- Meuthia Aula Jabbar performed the experiments, prepared figures and/or tables, and approved the final draft.
- Ramadhona Saville conceived and designed the experiments, authored or reviewed drafts of the article, and approved the final draft.
- Masaaki Wada conceived and designed the experiments, authored or reviewed drafts of the article, and approved the final draft.

### Human Ethics

The following information was supplied relating to ethical approvals (i.e., approving body and any reference numbers):

Research Ethics Committee on Social Studies and Humanities, National Research and Innovation Agency:

### Data Availability

The data is available at Zenodo: Manik, Lindung Parningotan. (2022). Dataset of Usability and Acceptance of Alboom: A Crowd-Based Early Warning System for Harmful Algal Blooms [Data set]. Zenodo. https://doi.org/10.5281/zenodo.7491272.

### Supplemental Information

Supplemental information for this article can be found online at http://dx.doi.org/10.7717/peerj.14923#supplemental-information.

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
