# Peer review of "Usability and acceptance of crowd-based early warning of harmful algal blooms"

_PeerJ, doi:10.7717/peerj.14923_

## Round 0.1 · original submission · Minor Revisions

The manuscript has been reviewed by two experts in this field. The authors need to address the points raised by both reviewers including aspects on anthropogenic stressors on HAB more clearly along with the uniqueness of HAB app which has been used in this study.

·

Basic reporting

Subsection 'Abloom' within section 'Literature Review' could be potentially re-written in terms of the functioning of Abloom.

Experimental design

no comment

Validity of the findings

no comments

Additional comments

This paper presents usability and acceptance analysis of a crowd-based early warning system (CBEWS) in an effort to solve the problem of Harmful Algal Blooms (HABs). Mainly using native technology acceptance model (TAM) covariables, this study applies the partial least square-structural modelling to understand and quantify the human response towards HABs in terms of awareness and willingness to contribute towards mitigation from a crowdsensing perspective. This study uses a proposed platform called 'Abloom' to test the same. It was reported that more than rewards, usability, problem awareness and tool usefulness contributed to people/citizens wanting to use 'Abloom'.

However, I feel that this article could benefit from a little more environmental perspective and a clear write-up about how Abloom functions in terms of detecting a HAB and what kind of components are involved.

Reviewer 2 ·

Basic reporting

English language is largely clear and unambiguous and the references are largely appropriate. Please sees comments below. The article is professionally structure and raw data is shared. Results are relevant to clearly outline hypotheses.

Experimental design

The manuscript is original research within the aims and scope of the journal. The research questions is largely will define except that stating that the application of the TAM in acceptance studies for crowd-based technologies is rare, is not a strong enough justification to the research. The application of a HAB app is ancillary and is not neatly related to the purpose of the manuscript. Please see comments below. The methods are clearly described and of high standard

Validity of the findings

Conclusions are well stated, linked to original research question & limited to supporting results. All underlying data have been provided; they are robust, statistically sound, & controlled

Additional comments

The manuscript, “Usability and acceptance of crowd-based early warning of harmful algal blooms,” examines the willingness of communities to be actively and continually involved in using crowd-based early warning systems.

In the literature review, the authors define crowdsourcing, the technological acceptance model, the concept of usability, HABs and Alboom, the mobile application which will be used as an early warning system for algal blooms. The authors indicate that the application of the Technology Acceptance Model is rare and state that the ease of use and usefulness are essential for crowdfunding projects. The application of TAMS to EWS is limited, so that authors set out to apply the TAM to assess usability and various other factors to EWS.

The authors clearly set out a to testable hypotheses based on a research model. The hypotheses revolve around various factors such as the ease of use and attitude and how they influence the use of CBEWS.

Data were collected with an online survey utilizing and a Likert scale to categorize responses. The survey has ethics clearance and includes keeping the respondents anonymous. The survey results were statistically analyze with SmartPLS using partial least squares and Cronbach’s alpha.

The results and the statistics are clearly summarized and are used to support the hypotheses that are tested.

Though I am not familiar with the TAM approach and SUS scores, the authors seem to clearly outline the process and involved in using this approach making it easily replicated. Figures and table are clear, though the captions are short and uninformative. More detail should be provided within the figure captions so that they can be used independently of the text of the manuscript.

The most important issue is that the concept of HABs is not addressed in a thorough manner. There are several types of HABs and the authors use the definition of biomass accumulation. Biomass accumulation is visible for reporting through the Alboom app. Toxic HABs are also another type of HAB, which are likely more detrimental to the aquaculture industry. The Alboom app would not be appropriate for the toxic type of HAB, as biomass accumulation is not typically associated with toxin accumulation and toxins are not visible. The authors should provide more background on the types of HABs found in the region along with several references. According the Keon-Munoz et al., (2018), the most widely accepted mechanism of these kills is the presence of an icthyotoxin, not biomass accumulation? How would the Alboom app work with crowd sensing toxic-type HABs?

Also, the authors state that HAB events (Line 136) tend to increase with increases in sea surface temperature. Please provide some references for this. This is not always the case, especially for toxin producing dinoflagellates.

The sentence on lines 56-57 is vague and unclear. Please rewrite. What do you mean by, “to obtain the desired information.”

Data file are provided; however, the supplemental files need more descriptive metadata identifiers to be useful to future readers. References seem to be in order.

---

## Round 0.2 · accepted · Accept

The manuscript is now accepted.

·

Basic reporting

Language and write-up are grammatically correct, clear and unambiguous.

Experimental design

Fit to publish

Validity of the findings

Fit to publish

Additional comments

I am happy with the revisions made by the authors. I think the article is fit for publishing at this point.